**Data Availability Statement:** All relevant data are within the paper and its Supporting Information files.

# Selected wetland soil properties correlate to Rift Valley fever livestock mortalities reported in 2009-10 in central South Africa

Anna M. Verster[1], Janice E. Liang[2], Melinda K. Rostal[2], Alan Kemp[3], Robert F. Brand [1,¤], Assaf Anyamba[4,5], Claudia Cordel[6], Robert Schall[7], Herman Zwiegers[1], Janusz T. Paweska[3], William B. Karesh[2], Cornie W. van Huyssteen [1] *

1 Department of Soil, Crop and Climate Sciences, University of the Free State, Bloemfontein, Republic of South Africa, 2 EcoHealth Alliance, New York City, New York, United States of America, 3 National Institute for Communicable Diseases, Johannesburg, Republic of South Africa, 4 NASA Goddard Space Flight Center, Greenbelt, Maryland, United States of America, 5 Universities Space Research Association, Columbia, Maryland, United States of America, 6 ExecuVet Veterinary Clinical and Scientific Consulting, Bloemfontein, Republic of South Africa, 7 Department of Mathematical Statistics and Actuarial Sciences, University of the Free State, Bloemfontein, Republic of South Africa

¤ Current address: Cuyahoga County Board of Health, Parma, Ohio, United States of America
* vanhuyssteencw@ufs.ac.za

## Abstract

Outbreaks of Rift Valley fever have devastating impacts on ruminants, humans, as well as on regional and national economies. Although numerous studies on the impact and outbreak of Rift Valley fever exist, relatively little is known about the role of environmental factors, especially soil, on the aestivation of the virus. This study thus selected 22 sites for study in central South Africa, known to be the recurrent epicenter of widespread Rift Valley fever outbreaks in Southern Africa. Soils were described, sampled and analyzed in detail at each site. Of all the soil variables analyzed for, only eight (cation exchange capacity, exchangeable $Ca^{2+}$, exchangeable $K^+$, exchangeable $Mg^{2+}$, soluble $Ca^{2+}$, medium sand, As, and Br) were statistically identified to be potential indicators of sites with reported Rift Valley fever mortalities, as reported for the 2009–2010 Rift Valley fever outbreak. Four soil characteristics (exchangeable $K^+$, exchangeable $Mg^{2+}$, medium sand, and Br) were subsequently included in a discriminant function that could potentially be used to predict sites that had reported Rift Valley fever-associated mortalities in livestock. This study therefore constitutes an initial attempt to predict sites prone to Rift Valley fever livestock mortality from soil properties and thus serves as a basis for broader research on the interaction between soil, mosquitoes and Rift Valley fever virus. Future research should include other environmental components such as vegetation, climate, and water properties as well as correlating soil properties with floodwater *Aedes* spp. abundance and Rift Valley fever virus prevalence.

**Funding:** WBK, MKR and JTP are thankful to the U. S. Department of Defense, Defense Threat Reduction Agency's Biological Threat Reduction Program for funding. Grant number: HDTRA1-14-1-0029. URL: https://www.dtra.mil/Mission/Mission- irectorates/Cooperative-Threat-Reduction/#. The funders had no role in study design, data collection and analysis, decision to publish, or preparation of the manuscript. The project depicted is sponsored by the U.S. Department of Defense, Defense Threat Reduction Agency. The content of the information does not necessarily reflect the position or the policy of the federal government, and no official endorsement should be inferred. Claudia Cordel is affiliated with ExecuVet Veterinary Clinical and Scientific Consulting a private consulting company hired through the funding received by the listed funders. ExecuVet Veterinary Clinical and Scientific Consulting provided support in the form of salaries for authors CC, but did not have any additional role in the study design, data collection and analysis, decision to publish, or preparation of the manuscript. The specific role of this author is articulated in the 'author contributions' section.

**Competing interests:** We have the following interests: Claudia Cordel is affiliated with ExecuVet Veterinary Clinical and Scientific Consulting. ExecuVet Veterinary Clinical and Scientific Consulting is a private company that was hired as a subcontractor to implement aspects of the research related to field data collection. They made no contributions toward funding this project. There are no patents, products in development or marketed products to declare. This does not alter our adherence to all the PLOS ONE policies on sharing data and materials.

## Introduction

Outbreaks of Rift Valley fever (RVF) have significant impacts on animal and human health as well as the economy [1, 2]. In livestock, the RVF virus (RVFV) can cause abortion in up to 100% of pregnant animals, as well as hepatic disease [3]. Livestock are infected with RVFV via the bite of an infected mosquito [3], whereas people are most frequently infected through direct and indirect contact with infected animal tissue, blood, or other bodily fluids [4]. Symptoms in humans are usually mild, although it may lead to death in a small proportion of cases [5]. Large outbreaks of RVF tend to be followed by years with very low levels of inter-epidemic transmission of RVFV and few diagnosed cases in people or animals [6].

Rift Valley fever virus is a vector-borne zoonosis and certain floodwater *Aedes* spp. and *Culex* spp. mosquitoes are implicated as important vectors [6]. It is believed that floodwater *Aedes* spp. mosquitoes can transmit the RVFV both horizontally and transovarially [7, 8]. The transovarial transmission may thus allow the maintenance of RVFV during the long inter-epidemic periods. *Aedes* spp. mosquitoes typically lay their eggs singly on the surface of the moist soil substrate, at the edge of the water surface [7, 9], in shallow grassland depressions, or in endorheic pans (also called *playas* or *dambos* in East Africa) [10]. These eggs desiccate and enter a state of dormancy that is broken only when heavy rainfall occurs and the flooded pan triggers hatching of the embryonated eggs [6, 11]. With the appropriate humidity, floodwater mosquito eggs can potentially remain quiescent in wetland soils for up to 12 months or longer [12, 13]. Understanding the factors that contribute to the survival of the *Aedes* spp. eggs could therefore improve our ability to predict when and where RVFV outbreaks are likely to occur.

RVF outbreaks are highly correlated with anomalous high rainfall, and subsequent flooding that allows for dormant *Aedes* spp. eggs to hatch [14, 15, 16]. Despite an accurate prediction of the 2006–2007 RVF outbreak in East Africa, rainfall anomalies and rapid increase in normalized difference vegetation index (NDVI) have, however, not always been able to accurately predict epidemics. This indicates that there are other environmental or host factors that contribute to the temporal pattern of RVF outbreaks [16, 17]. Only one study was found that developed a RVF risk map, using rainfall and soil water saturation levels at the landscape scale [18]. Additionally, Hightower et al. [19] and Sindato et al. [20] suggest that RVF outbreaks were highly associated with Solonetz, Calcisol, Solonchak, and Planosol soil types and with soils with low water permeability. These associations between soil type, soil moisture and epidemics are large-scale observations and do not suggest mechanisms for linking specific ecological factors to RVFV epidemics.

We hypothesize that soil properties, including moisture, are associated with sites where *Aedes* spp. mosquito eggs and RVFV have higher levels of survival, which ultimately manifests as RVF mortality in livestock during outbreaks. To test this hypothesis we compared the soil characteristics of wetlands where RVF livestock mortalities were reported to the soil characteristics of wetlands where no mortalities were reported during the 2009–2010 RVF outbreak in the Free State and Northern Cape Provinces of South Africa. By understanding the ecology of RVFV, we thus hope to improve predictive and mitigation measures to reduce the impact of future RVF outbreaks.

## Material and methods

### Study area

Within South Africa, the highest livestock mortalities, associated with 2009–2010 RVF outbreak have primarily occurred in the Free State and Northern Cape Provinces (Fig 1). This study therefore focused on these provinces.

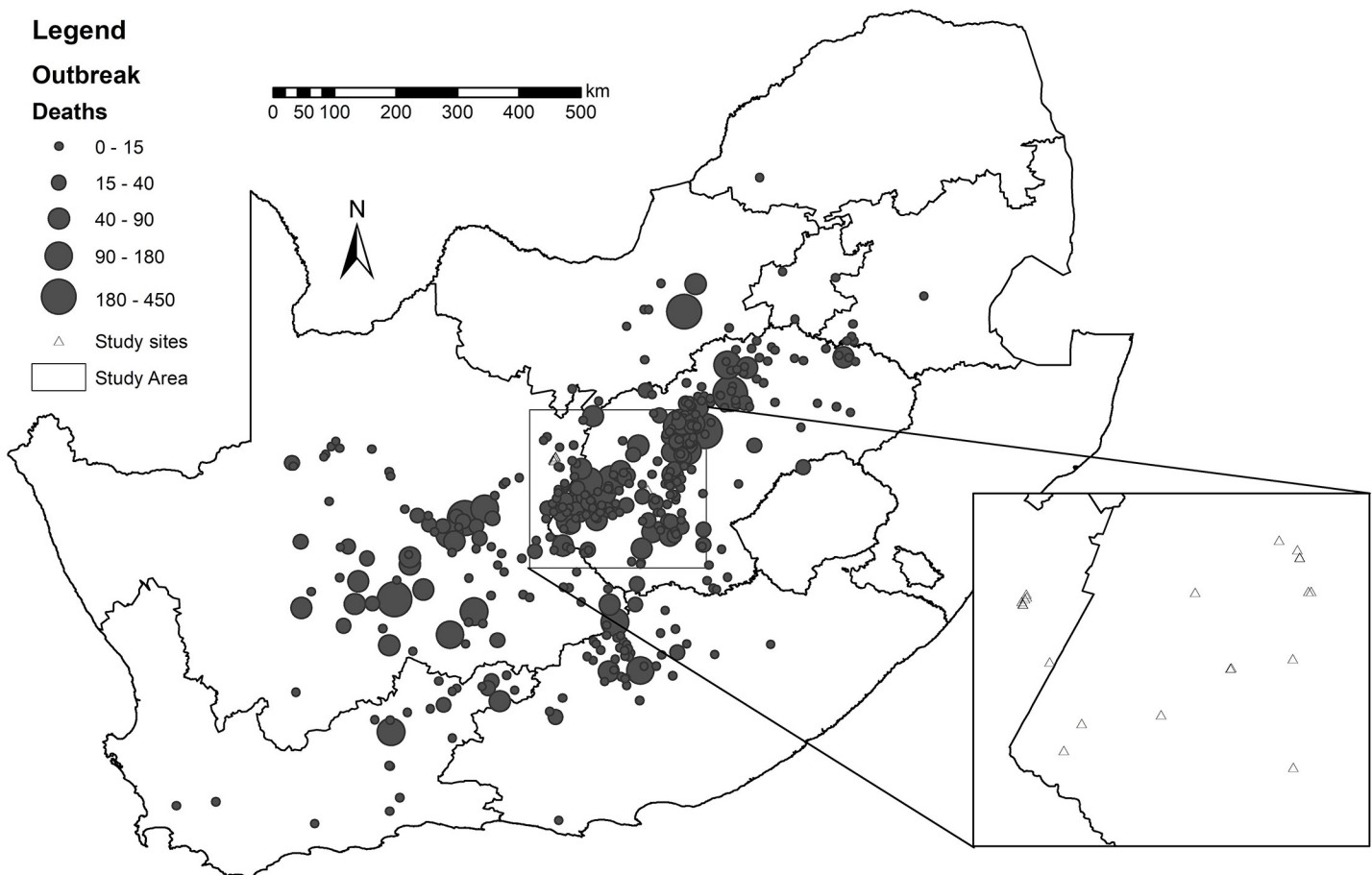

**Fig 1. Sheep mortalities (number of reported deaths) associated with the 2009–2010 Rift Valley fever outbreak in South Africa (map compiled using data from OIE Follow-Up Report [21]).**

### Study site selection

An area of approximately 200 km by 200 km in the Free State and Northern Cape Provinces was chosen for this study (Fig 1). Five sites were selected at approximately 40 km intervals along an east-west transect from Bloemfontein to the Mokala National Park, also representing a moisture gradient, regardless of whether or not RVF livestock mortality had been reported to the OIE in 2009–2010 [21]. Ten sites were additionally selected based on locations with reported mortalities in livestock, while an additional seven sites were selected that did not report livestock mortalities, giving 22 sites in total. The field work was primarily conducted on private land and the owners gave us explicit permission to conduct the research on their land. One site was managed by the South African National Parks and for this site we worked under permit: KAREW1257. Study sites were subsequently grouped into those where RVF livestock mortality had been reported and those where no livestock mortality had been reported [21]. The site ID, closest town, latitude, longitude, and dominant soil type are given in Table 1.

### Soil sampling

Soil samples were collected in May and June 2015, using a soil auger. Soil samples were collected from the permanent wet zone of the wetland [22], with plot sizes varying (approximately 10 m$^2$) according to wetland type, location and physical access to sites. These plots were also

**Table 1. Site ID, site name, closest town, coordinates, and World Reference Base (WRB) soil classification [26] of the wetland sites where RVF mortalities have and have not been reported [21].**

|  | Site ID | Town | Latitude | Longitude | WRB Soil Classification |
|---|---|---|---|---|---|
| Sites with reported RVF livestock mortalities | p013B | Brandfort | -28.628 | 26.316 | Calcic Kastanozem (Loamic, Cambic, Stagnic) |
|  | p001B | Brandfort | -28.628 | 26.316 | Calcic Kastanozem (Anthric, Clayic, Pachic, Stagnic) |
|  | p002B | Bultfontein | -28.400 | 26.258 | Haplic Vertisol (Mollic, Stagnic) |
|  | p004B | Bultfontein | -28.404 | 26.119 | Protostagnic Kastanozem (Loamic, Stagnic) |
|  | p009D | Dealesville | -28.639 | 25.572 | Haplic Calcisol (Clayic, Hypercalcic, Stagnic) |
|  | p009D_2 | Dealesville | -28.635 | 25.576 | Hypocalcic Kastanozem (Clayic) |
|  | p011P | Petrusburg | -29.140 | 25.375 | Hypocalcic Kastanozem (Clayic) |
|  | p010J | Jacobsdal | -29.092 | 24.607 | Haplic Calcisol (Loamic, Hypercalcic) |
|  | p008O | Koffiefontein | -29.492 | 24.837 | Haplic Calcisol (Clayic, Hypocalcic, Stagnic) |
|  | p007L | Luckhoff | -29.672 | 24.701 | Haplic Calcisol (Arenic, Hypercalcic, Stagnic) |
|  | p005P | Koffiefontein | -29.438 | 25.351 | Hypocalcic Kastanozem (Loamic, Cambic, Oxyaquic) |
|  | p012R | Reddersburg | -29.780 | 26.223 | Haplic Kastanozem (Clayic, Chromic) |
| Sites with no reported RVF livestock mortalities | p003B | Bultfontein | -28.352 | 26.246 | Calcic Mollic Stagnosol (Loamic) |
|  | p014B | Bloemfontein | -29.069 | 26.216 | Cambic Chernic Umbrisol (Loamic) |
|  | p006B | De Brug | -29.136 | 25.800 | Haplic Umbrisol (Clayic, Pachic) |
|  | p006B_2 | De Brug | -29.124 | 25.807 | Haplic Calcisol (Loamic, Hypercalcic) |
|  | p015K | Kimberley | -28.695 | 24.428 | Haplic Stagnosol (Loamic) |
|  | p015K_4 | Kimberley | -28.675 | 24.458 | Haplic Calcisol (Loamic, Hypercalcic) |
|  | p015K_5 | Kimberley | -28.675 | 24.448 | Haplic Calcisol (Loamic, Hypercalcic, Protostagnic) |
|  | p015K_2 | Kimberley | -28.708 | 24.434 | Haplic Calcisol (Loamic, Hypercalcic) |
|  | p015K_3 | Kimberley | -28.712 | 24.440 | Haplic Calcisol (Loamic, Hypercalcic) |
|  | p015K_6 | Kimberley | -28.661 | 24.459 | Haplic Calcisol (Loamic, Hypocalcic) |

investigated for vegetation [23], mosquito and weather-related factors, as part of a larger study. Composite soil samples, consisting of sub-samples from six auger holes within each plot were collected. The six sub-samples were mixed thoroughly, and a representative sample of approximately 1 kg was taken for further analysis. Soil samples were collected from the surface layer (0–50 mm) and from each dominant soil layer, also termed the master horizon [24], yielding 3 to 4 sampled layers per site. A detailed soil profile description [25] and soil profile classification, using IUSS Working Group WRB [26], was done at each plot.

A surface organic material (dead vegetation) sample was also collected from a 1 m$^2$ area within each plot for active carbon analysis. Each 1 m$^2$ area was divided into four quarters. An approximately 100 g sub-sample was collected from each quarter, sieved through a 2 mm mesh and recombined, from which a representative sample of approximately 100 g was collected for further analysis. Latex gloves were used during the collection of the surface organic material to avoid sample contamination.

Soil and surface organic material samples were analyzed from June 2015 to January 2016 at the University of the Free State, Bloemfontein.

## Soil analyses

Soil physical and chemical analyses of all sampled soil layers were completed using standard procedures [27]. Texture was analyzed by determining the sand, silt, and clay content of the soil in seven fractions. Chemical analyses included organic carbon, total nitrogen (dry oxidation), pH (water and KCl), electrical resistance, soluble and exchangeable cations, and cation exchange capacity (CEC).

The soil clay fraction mineralogy was characterized using X-ray diffraction (XRD) analysis [28]. XRD patterns were obtained with an Empyrean theta-theta diffractometer (Malvern-Panalytical, Netherlands) equipped with a copper anode X-ray tube, operating at 45 kV and 40 mA. The measurements were carried out in Bragg-Brentano mode according to the manufacturer's instructions. Phase identification and semi-quantitative analyses were performed using the Highscore software (Malvern-Panalytical, Netherlands). The elemental composition of the <2 mm soil fraction was determined using X-ray fluorescence (XRF) [29]. An Axios XRF spectrometer (Malvern-Panalytical, Netherlands) with a 4 kW anode and a 1 W cathode was used. Both XRD and XRF determinations were performed by the Department of Geology at the University of the Free State. Mineralogical analyses were done for the samples from the 0–50 mm layer only, since this is the layer where floodwater *Aedes* spp. mosquitoes would lay their eggs [11].

Total microbial activity (active fungi, bacteria, and protists present) of the surface organic material samples were analyzed at each site [30]. Before analysis, 2 g samples were prepared and stored in 50 ml centrifuge tubes in a refrigerator (at 4˚C). Fluorescein diacetate (FDA) hydrolysis [31] was performed using methods outlined by Adam and Duncan [30], as modified by Zabaloy et al. [32]. The 2 g soil sample was mixed with a 200 ml, 60 mM sodium phosphate buffer, at a pH of 7.6, in a 50 ml plastic centrifuge tube. Then 0.2 ml of the 2 mg/ml FDA stock solution was added [31]. The FDA stock solution was prepared by adding 200 mg FDA to 100 ml acetone and stored at -20˚C until it was used to determine the microbial activity. The 50 ml tubes containing the samples were incubated for 20 minutes at 28˚C in a Labcon growth chamber (Air & Vacuum Technologies, South Africa). During the incubation period, the tubes were shaken manually three times. After the incubation period, the tubes were placed on a Multi Reax shaker (Heidolph, Germany) at 300 rpm for 10 minutes at room temperature. To end the hydrolysis reaction, 15 ml of a 2:1 chloroform methanol solution was added. Approximately 1.5 ml of the supernatant was then placed in an Eppendorf tube (Hamburg, Germany) and centrifuged at 200 rpm for 3 minutes. An aliquot was subsequently placed in a microtiter plate and absorbance measurements were taken at 490 nm. For each sample, two controls were also prepared. The first control consisted of 2 g of sample and the phosphate buffer, without any FDA stock solution added. The second control contained the phosphate buffer and FDA stock solution without any sample.

Active carbon or permanganate oxidizable carbon (C) was analyzed according to Culman et al. [33]. A of sieved air-dried soil sample of 2.5 g was placed into 50 ml tubes, wherein 18 ml of deionized water and 2 ml of 0.02 M $KMnO_2$ solution were added. The tubes were shaken for 2 minutes at 240 rpm on an oscillating shaker and then centrifuged for 5 minutes at 3000 rpm. A volume of 0.5 ml of the supernatant was immediately transferred to new 50 ml tubes and diluted with 49.5 ml deionized water before measuring the absorbance on a spectrophotometer at 550 nm. The amount of carbon oxidized was then calculated as a function of the quantity of the measured reduced permanganate. The final active carbon was calculated using the equation proposed by Weil et al. [34].

## Statistical analysis

Statistical analyses were performed using SAS software [35] and included descriptive statistics, between-group comparisons, Pearson correlation coefficients, and discriminant analysis. For the chemical and physical soil properties, only the top three layers (0–50 mm, A horizon, B/B1/G/C1 horizon) were used in the statistical analyses since the lowest layer was not collected at all sites. For the mineralogical, microbiology, and active carbon analysis, only the one sampled 0–50 mm layer was used in statistical analyses.

The various chemical and physical soil properties were compared between the two groups of sites (reported versus not reported RVF mortalities) using the non-parametric van der Waerden test [35]. This test if the equality of population means remains valid for data that are not normally distributed [35]. The comparisons were conducted per layer and for the average of the first three layers for chemical and physical analyses, and for the single sampled layer for mineralogical, microbiology, and active carbon analyses.

Discriminant analysis [35] was conducted as follows: For each of the soil properties selected as significant or near significant through the van der Waerden test, the optimal transformation to normality was determined using the Box-Cox method for each of the first three layers as implemented in the SAS procedure "TRANSREG". Thus, the optimal parameter of the power transformation was determined; thereafter the overall best parameter across the three layers was chosen by inspection (based on likelihood profiles as a function of the power parameter). For each site, the mean of the first three layers for each of the transformed variables was used (except for the mineralogical and microbiological variables since only one layer was assayed). Stepwise selection for discriminant analysis of the above transformed (and averaged) variables was carried out using the SAS procedure "STEPDISC". Variables were kept in the discriminant analysis based on a 0.1 significance level. Quadratic discriminant analysis was carried out using the variables included in the model using the SAS procedure "DISCRIM." The misclassi-fication rate of the quadratic discriminant function was estimated using cross-validation.

## Results

### Between-group comparisons

The mean, standard deviation, median, minimum, and maximum for each measured variable were calculated by layer (S1 Table; S2 Table) and group of sites (RVF mortalities reported ver-sus not reported; Table 2; Table 3). Between-group comparisons of the means for chemical and physical soil properties of the first three layers and the means for microbiology and miner-alogical properties were used to identify potential predictors of sites with and without reported RVFV mortalities. Tables 2 and 3 give the results of the non-parametric test and highlight the characteristics and soil layers that were significantly different between the sites with reported RVF mortalities.

The non-parametric test identified a difference between mean values of the sites with reported RVF mortalities and sites without as statistically significant (p-value <0.1) for the fol-lowing properties: soluble $Ca^{2+}$, exchangeable $Ca^{2+}$, exchangeable $Mg^{2+}$, exchangeable $K^+$, CEC, and medium sand (Table 2) as well as As and Br (Table 3). These soil properties could therefore possibly be used as predictors of locations where previous RVF mortality was reported. Fig 2 illustrates the descriptive statistics of the eight variables that were included in the discriminant analysis, while Table 4 presents the pairwise Pearson correlation coefficients between these variables. All of the mean values for eight of these variables were lower at the sites with reported RVF mortalities than for the sites that had not.

The transformations identified by the Box-Cox analysis were as follows: square root for CEC, no transformation for exchangeable $Ca^{2+}$, natural logarithm for soluble $Ca^{2+}$, square root for exchangeable $K^+$, square root for medium sand, logarithm for exchangeable $Mg^{2+}$, square root for As, and logarithm for Br.

### Linear discriminant function

Of the eight variables selected for further analysis after the initial between-group comparisons, four variables (exchangeable $K^+$, exchangeable $Mg^{2+}$, medium sand, and Br) were identified by the stepwise variable selection of the discriminant analysis to discriminate between sites with

**Table 2. Means for the chemical and physical soil properties across all sites where RVF mortalities were reported or not reported [21] during the 2009–2010 RVF outbreak (calculated using the mean values per site of the first three soil layers).** Significance was determined using the non-parametric van der Waerden test, and is given as the p-value.

| Variable | Group (RVF mortalities) | | p-value |
|---|---|---|---|
| | Reported | Not reported | |
| Soluble $Ca^{2+}$ ($cmol_c$ $kg^{-1}$) | **0.17** | **0.49** | **0.0005** |
| Soluble $Mg^{2+}$ ($cmol_c$ $kg^{-1}$) | 0.17 | 0.14 | 0.9212 |
| Soluble $K^+$ ($cmol_c$ $kg^{-1}$) | 0.01 | 0.01 | 0.3013 |
| Soluble $Na^+$ ($cmol_c$ $kg^{-1}$) | 2.95 | 1.44 | 0.4845 |
| Exch. $Ca^{2+}$ ($cmol_c$ $kg^{-1}$) | **25.5** | **45.5** | **0.0016** |
| Exch. $Mg^{2+}$ ($cmol_c$ $kg^{-1}$) | **7.27** | **11.2** | **0.0231** |
| Exch. $K^+$ ($cmol_c$ $kg^{-1}$) | **1.69** | **3.01** | **0.0033** |
| Exch. $Na^+$ ($cmol_c$ $kg^{-1}$) | 2.09 | 7.65 | 0.7162 |
| CEC ($cmol_c$ $kg^{-1}$) | **16.7** | **28.1** | **0.0065** |
| Organic carbon (mg $kg^{-1}$) | 28825 | 26926 | 0.9909 |
| Total nitrogen (mg $kg^{-1}$) | 1465 | 1478 | 0.8049 |
| Electrical resistance ($\Omega$) | 1465 | 14778 | 0.8049 |
| $pH_{KCl}$ | 7.27 | 7.19 | 0.9087 |
| $pH_{Water}$ | 8.26 | 8.14 | 0.7173 |
| Coarse sand (%) | 3.6 | 4.98 | 0.4089 |
| Medium sand (%) | **4.71** | **7.20** | **0.0349** |
| Fine sand (%) | 21.9 | 26.3 | 0.1695 |
| Very fine sand (%) | 13.1 | 13.3 | 0.3809 |
| Coarse silt (%) | 6.65 | 6.72 | 0.7192 |
| Fine silt (%) | 14.6 | 13.4 | 0.8343 |
| Clay (%) | 33.1 | 24.0 | 0.1683 |
| Residual silt and clay (%) | 1.72 | 2.11 | 0.1725 |

Values in bold-type differ significantly at p<0.1

and without reported RVF mortality. The average misclassification rate (4.2%) was estimated through cross-validation. The following linear discriminant function could thus be used to predict if a site had or did not have reported RVF mortalities:

$$D_g(x_i) = x_i' L_g + C_g$$

Here, for a given wetland $i$, $x_i$ is the vector of four dimensions containing the values of the variables: sqrt(exchangeable $K^+$), ln(exchangeable $Mg^{2+}$), sqrt(medium sand) and ln(Br).

The vectors, $L_g$ and constants $C_g$ were as follows (outputs of the SAS procedure "DISCRIM"), where g = group:

$$L_1 = \begin{pmatrix} 29.828 \\ 14.982 \\ 7.825 \\ 1.318 \end{pmatrix}$$

$$C_1 = -42.773$$

**Table 3. Means of the soil microbiology, mineralogical, and elemental properties within the sampled layer across all sites where RVF mortalities were reported or not reported [21] during the 2009–2010 RVF outbreak.** Significance was determined using the non-parametric van der Waerden test, and is given as the p-value.

| Variable | Group (RVF mortalities) | | |
| --- | --- | --- | --- |
| | Reported | Not reported | p-value |
| Active Carbon (mg kg$^{-1}$) | 482 | 411 | 0.2518 |
| FDA (µg FDA/g soil) | 89.1 | 57.1 | 0.4407 |
| Anatase (% m/m) | 0.47 | 0.39 | 1.0000 |
| Andalusite (% m/m) | 0.74 | 0 | 1.0000 |
| Ankerite (% m/m) | 1.84 | 4 | 0.4135 |
| Apophyllite (% m/m) | 1.04 | 0 | 1.0000 |
| Calcite (% m/m) | 4.52 | 4.82 | 0.9137 |
| Dolomite (% m/m) | 2.84 | 2.19 | 0.8209 |
| Gypsum (% m/m) | 1.33 | 0 | 1.0000 |
| Halite (% m/m) | 4.16 | 3.98 | 0.8587 |
| K-feldspar/rutile (% m/m) | 8.6 | 8.83 | 0.7361 |
| Kaolinite (% m/m) | 2.69 | 4.92 | 0.5414 |
| Mica (% m/m) | 24.5 | 24.5 | 0.8600 |
| Plagioclase (% m/m) | 18.1 | 14.6 | 0.1291 |
| Pyroxene (% m/m) | 0 | 3.19 | 0.1948 |
| Quartz (% m/m) | 27 | 25.7 | 0.7030 |
| Smectite (% m/m) | 2.12 | 2.84 | 0.7403 |
| Al$_2$O$_3$ (% m/m) | 7.93 | 8.89 | 0.2890 |
| CaO (% m/m) | 4.81 | 3.53 | 0.6578 |
| Fe$_2$O$_3$ (% m/m) | 4.23 | 4.96 | 0.2893 |
| K$_2$O (% m/m) | 1.43 | 1.58 | 0.4668 |
| MgO (% m/m) | 2.3 | 2.57 | 0.9354 |
| MnO (% m/m) | 0.05 | 0.08 | 0.2834 |
| Na$_2$O (% m/m) | 1 | 0.61 | 0.3060 |
| P$_2$O$_5$ (% m/m) | 0.09 | 0.11 | 0.6942 |
| SiO$_2$ (% m/m) | 64.4 | 66.1 | 0.4367 |
| TiO$_2$ (% m/m) | 0.49 | 0.5 | 0.8574 |
| As (mg kg$^{-1}$) | **8.49** | **5.96** | **0.0999** |
| Ba (mg kg$^{-1}$) | 748 | 667 | 0.2872 |
| Br (mg kg$^{-1}$) | **29.9** | **11** | **0.0764** |
| Co (mg kg$^{-1}$) | 10.9 | 13.8 | 0.4069 |
| Cr (mg kg$^{-1}$) | 94 | 91.9 | 0.9646 |
| Cu (mg kg$^{-1}$) | 30 | 28.3 | 0.8139 |
| Nb (mg kg$^{-1}$) | 3.81 | 3.77 | 0.9529 |
| Ni (mg kg$^{-1}$) | 31.5 | 38.6 | 0.1312 |
| Pb (mg kg$^{-1}$) | 10.6 | 11.9 | 0.7141 |
| Rb (mg kg$^{-1}$) | 61.7 | 61.6 | 0.6061 |
| Sc (mg kg$^{-1}$) | 9.49 | 7.6 | 0.3919 |
| Sr (mg kg$^{-1}$) | 332 | 177 | 0.9497 |
| Th (mg kg$^{-1}$) | 3.3 | 2.64 | 0.7571 |
| V (mg kg$^{-1}$) | 109 | 91.9 | 0.3076 |
| Y (mg kg$^{-1}$) | 15.4 | 16.7 | 0.5581 |
| Zn (mg kg$^{-1}$) | 49 | 50.6 | 0.9509 |
| Zr (mg kg$^{-1}$) | 208 | 204 | 0.7144 |

Values in bold-type differ significantly at p <0.1

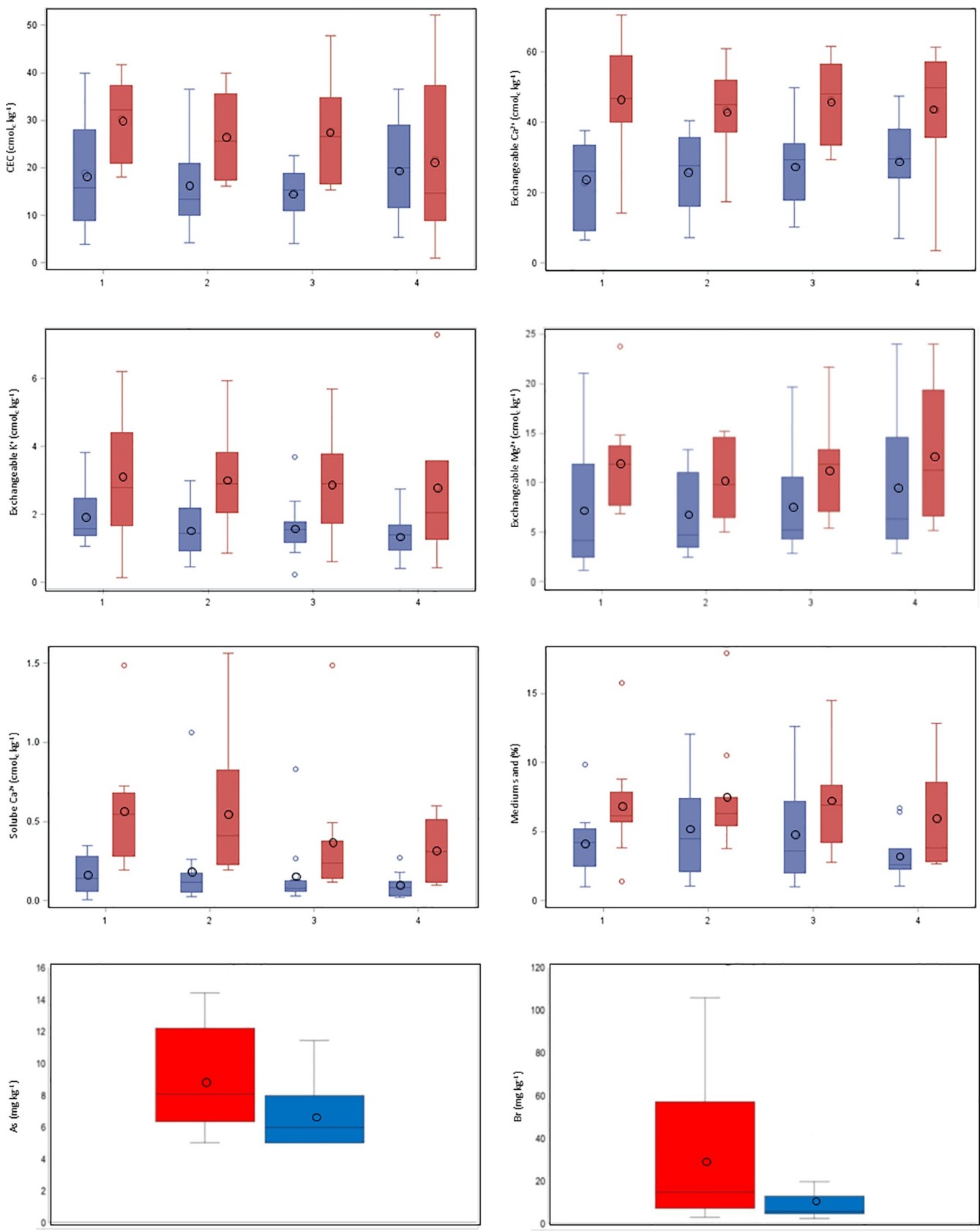

Fig 2. Box-and-whisker plots denoting the descriptive statistics for CEC, exchangeable Ca²⁺, exchangeable K⁺, exchangeable Mg²⁺, soluble Ca²⁺, medium sand, As, and Br, included in the discriminant analysis. Boxes indicate the standard deviations, whiskers the minimums and maximums, black circles the means, and horizontal lines the median. On the X-axis 1 = 0–50 mm layers; 2 = A horizons; 3 = B/B1/G/C1 horizons; 4 = B2/C/C2 horizons. Blue boxes represent sites with, while red boxes represent sites with no reported RVF mortalities [21]. Colored circles indicate extraneous values that were deleted.

$$L_2 = \begin{pmatrix} 38.720 \\ 19.937 \\ 11.943 \\ -0.225 \end{pmatrix}$$

$$C_2 = -71.929$$

For each location $i$, the distances $D_g(x_i)$ of $x_i$ from the two groups, g = 1, 2 are calculated and then location $i$ is allocated to Group 1 if $D_1(x_i) < D_2(x_i)$ otherwise, if $D_1(x_i) > D_2(x_i)$, location $i$ is allocated to Group 2.

## Discussion

The soil properties identified above constitute an initial attempt to predict sites prone to RVF livestock mortality using four primary soil characteristics: exchangeable K⁺, exchangeable Mg²⁺, medium sand, and Br.

The mean and range of medium sand was lower at sites with reported RVF mortalities than at sites without reported RVF mortalities. A recent study by Brand *et al.* [23] conducted a visual field assessment of soil type and found that high clay-content soils was a common factor in areas of high RVF mortality in the Free State and Northern Cape of South Africa. While clay was not found to be a significant factor in our study, a lower medium sand content may indicate there is higher water retention. Sindato et al. [36] recently determined that in northern and central-eastern Tanzania, soil type (impermeable soils) and rainfall in the wettest quarter of the year accounted for nearly two-thirds of the variation in habitat suitability for RVF occurrence. Impermeable soils, which are characterized by high clay content and loamy texture, do not allow water to easily filter through. This supports long periods of water retention, which contributes to flooding and wetness and thus provides a habitat suitable for the breeding, survival, and hatching of RVFV infected *Aedes spp*. mosquito eggs [37]. In contrast, permeable soils are characterized by sandy textures that do not result in as much flooding.

Although only exchangeable K⁺, exchangeable Mg²⁺, medium sand, and Br were used in the final model, the correlation between CEC on the one hand, and exchangeable Ca²⁺,

**Table 4. Pearson correlation coefficients of the eight variables found to be potential indicators of a site with reported RVF mortalities.**

|  | CEC | Exchangeable Ca²⁺ | Exchangeable K⁺ | Exchangeable Mg²⁺ | Soluble Ca²⁺ | Medium sand | As | Br |
|---|---|---|---|---|---|---|---|---|
| CEC | 1.00 |  |  |  |  |  |  |  |
| Exchangeable Ca²⁺ | 0.58 | 1.00 |  |  |  |  |  |  |
| Exchangeable K⁺ | 0.58 | 0.67 | 1.00 |  |  |  |  |  |
| Exchangeable Mg²⁺ | 0.75 | 0.63 | 0.39 | 1.00 |  |  |  |  |
| Soluble Ca²⁺ | 0.60 | 0.85 | 0.63 | 0.79 | 1.00 |  |  |  |
| Medium_sand | 0.05 | 0.17 | -0.24 | 0.07 | 0.05 | 1.00 |  |  |
| As | -0.21 | -0.16 | 0.05 | -0.16 | 0.03 | -0.41 | 1.00 |  |
| Br | -0.31 | -0.19 | -0.28 | 0.03 | -0.18 | -0.07 | 0.21 | 1.00 |

exchangeable $K^+$, exchangeable $Mg^{2+}$, and soluble $Ca^{2+}$ on the other hand, was relatively high (Table 4) and therefore these latter variables were not considered to be independent predictors of whether or not RVF mortalities had been reported. CEC represents the total amount of exchangeable cations that the soil can adsorb, and thus CEC and the basic cations ($Ca^{2+}$, $Mg^{2+}$, $Na^+$, $K^+$) frequently have a close relationship, with a higher CEC in the soil correlating to a higher basic cation concentration [38]. Additionally, CEC is also dependent on the clay content, clay type, organic matter content, and pH of the soil [38]. Therefore, a higher CEC (and in parallel, higher amounts of exchangeable $Ca^{2+}$, exchangeable $K^+$, and exchangeable $Mg^{2+}$) likely represents a more clayey, alkaline soil. The higher CEC and exchangeable cations in sites with observed RVF relates somewhat to the findings of Hightower et al. [19], that RFV is accociated with Solonetz, Calcisol, Solonchak, and Planosol soil types. Although not stated in their paper, these soils are normally associated with more clayey, alkaline environments.

The properties of soil influence the flooding and drainage of a wetland, and, potentially, the capacity for *Aedes* mosquito eggs to survive and remain vital in the soil until heavy rainfall occurs [39]. By identifying sites of past RVF outbreaks in South Africa, this study was able to retrospectively identify possible links between soil chemistry and mineralogy, microbiology, and RVF mortalities. Wetland soil characteristics could thus be used to improve predictive ecological models for identifying areas that are likely to be conducive to mosquito vector breeding and survival, and subsequent RVF outbreaks. Water content, pH, salinity, temperature, dissolved oxygen content, and lithology also merits investigation to enhance understanding of the requirements for vector production, in addition to the factors highlighted here. Common factors that have been related to areas of high RVF mortality in South Africa include the presence of low-salinity, freshwater and wetland vegetation composed mainly of sedges, *Juncus* and grasses [23].

By including other environmental components drawn from soil, geology, hydrology, vegetation and climate, a more accurate conclusion could possibly be drawn about the areas of southern Africa at greatest risk for future outbreaks of RVF. Predictive ecological/epidemiological models, based on these linkages are vital to provide insight on how best to mitigate and control future RVF outbreaks. In addition, future national soil surveys should include characterization of soil properties such as those highlighted here to serve as a reference for areas of potential RVF activity.

## Conclusions

A relatively large body of research exists on the relationships between environmental factors such as soil and water on a number of viruses (such as poliovirus, Coxsackie-virus, and reovirus) that affect human and animal health [40, 41, 42, 43]. However, research regarding the link between RVFV and environmental factors, such as soil, is minimal. Intensive research is therefore needed on how various soil elements can affect the microenvironment for the survival of RVFV and its vectors. Through statistical analysis, this study identified eight soil characteristics (CEC, exchangeable $Ca^{2+}$, exchangeable $K^+$, exchangeable $Mg^{2+}$, soluble $Ca^{2+}$, medium sand, As, and Br) to be potential indicators of sites with reported RVF mortalities. Four soil characteristics (exchangeable $K^+$, exchangeable $Mg^{2+}$, medium sand, and Br) were subsequently selected for inclusion into a discriminant function that could potentially be used to predict an RVF outbreak site. This study serves as a basis for broader scientific research on the interaction between soil, *Aedes* spp. mosquitoes and RVFV. Future work should focus on including other environmental components such as lithology, vegetation, climate, and water properties as well as correlating these soil properties with floodwater *Aedes* spp. abundance and RVFV prevalence. Understanding the complex connections between the various physical

and biotic variables, and vector ecology may mitigate the devastating effects of RVFV on animal and human health, and the local and national economies in regions at risk of these outbreaks.

## Supporting information

**S1 Table. Descriptive statistics of the physical and chemical properties by soil layer(where the "Reported" group consists of sites where RVF mortalities have been reported and "Not Reported" consists of sites where RVF mortalities have not been reported).**
(DOCX)

**S2 Table. Descriptive statistics of the soil microbiology and mineralogical analyses (where the "Reported" group are sites where RVF mortalities have been reported and the "Not Reported" group are sites where RVF mortalities have not been reported).**
(DOCX)

## Acknowledgments

The project depicted is sponsored by the U.S. Department of Defense, Defense Threat Reduction Agency. The content of the information does not necessarily reflect the position or the policy of the federal government, and no official endorsement should be inferred. The authors would also like to acknowledge the staff at ExecuVet for their logistical and in-field support.

## Author Contributions

**Conceptualization:** Melinda K. Rostal, Alan Kemp, Robert F. Brand, Assaf Anyamba, Janusz T. Paweska, William B. Karesh, Cornie W. van Huyssteen.

**Data curation:** Cornie W. van Huyssteen.

**Formal analysis:** Robert Schall.

**Funding acquisition:** Melinda K. Rostal, Janusz T. Paweska, William B. Karesh, Cornie W. van Huyssteen.

**Investigation:** Anna M. Verster, Claudia Cordel, Herman Zwiegers.

**Methodology:** Anna M. Verster, Robert F. Brand.

**Project administration:** Janice E. Liang, Melinda K. Rostal, Alan Kemp, Janusz T. Paweska, William B. Karesh, Cornie W. van Huyssteen.

**Resources:** Claudia Cordel, Cornie W. van Huyssteen.

**Supervision:** Cornie W. van Huyssteen.

**Validation:** Cornie W. van Huyssteen.

**Writing – original draft:** Anna M. Verster, Janice E. Liang, Melinda K. Rostal.

**Writing – review & editing:** Anna M. Verster, Janice E. Liang, Melinda K. Rostal, Alan Kemp, Robert F. Brand, Assaf Anyamba, Claudia Cordel, Robert Schall, Herman Zwiegers, Janusz T. Paweska, William B. Karesh, Cornie W. van Huyssteen.

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
