## [Decision Letter · Decision Letter 0]

27 Nov 2019

PONE-D-19-27989

Selected wetland soil properties correlate to Rift Valley fever livestock mortalities reported in 2009-10 in central South Africa

PLOS ONE

Dear Prof van Huyssteen,

Thank you for submitting your manuscript to PLOS ONE. After careful consideration, we feel that it has merit but does not fully meet PLOS ONE’s publication criteria as it currently stands. Therefore, we invite you to submit a revised version of the manuscript that addresses the points raised during the review process.

In particular one reviewer felt strongly that there were discrepancies in the main conclusions as to what affected the Rift Valley Fever mortalities. However, I feel that all comments should be addressed to the best of your ability.

We would appreciate receiving your revised manuscript by Jan 11 2020 11:59PM. To enhance the reproducibility of your results, we recommend that if applicable you deposit your laboratory protocols in protocols.io, where a protocol can be assigned its own identifier (DOI) such that it can be cited independently in the future. For instructions see: http://journals.plos.org/plosone/s/submission-guidelines#loc-laboratory-protocols

We look forward to receiving your revised manuscript.

Kind regards,

Naomi Forrester, PhD

Academic Editor

PLOS ONE

Journal Requirements:

'WBK, MKR and JTP are thankful to the U.S. Department of Defense, Defense Threat Reduction Agency's Biological Threat Reduction Program for funding. Grant number: HDTRA1-14-1-0029. URL: https://www.dtra.mil/Mission/Mission-Directorates/Cooperative-Threat-Reduction/#. The funders had no role in study design, data collection and analysis, decision to publish, or preparation of the manuscript. The project depicted is sponsored by the U.S. Department of Defense, Defense Threat Reduction Agency.  The content of the information does not necessarily reflect the position or the policy of the federal government, and no official endorsement should be inferred.'

We note that one or more of the authors are employed by a commercial company: nExecuVet Veterinary Clinical and Scientific Consulting.

Additional Editor Comments (if provided):

Reviewers' comments:

Reviewer's Responses to Questions

**Comments to the Author**

1. Is the manuscript technically sound, and do the data support the conclusions?

Reviewer #1: Yes

Reviewer #2: Partly

2. Has the statistical analysis been performed appropriately and rigorously? 

Reviewer #1: Yes

Reviewer #2: No

3. Have the authors made all data underlying the findings in their manuscript fully available?

Reviewer #1: Yes

Reviewer #2: Yes

4. Is the manuscript presented in an intelligible fashion and written in standard English?

Reviewer #1: Yes

Reviewer #2: Yes

5. Review Comments to the Author

Reviewer #1: Thank you for the opportunity to review this interesting research.

Overall the manuscript is well written and scientifically sound. I have mentioned to the editor that I do not have the appropriate expertise to comment on your selection of techniques for soil analysis.

I have recommended the manuscript undergo minor revisions which I have outlined below

Minor revisions

Abstract

1. CEC should be written in full in the first instance

2. I am not familiar with the standard nomenclature but should the cations also be written in full at the first mention

3. The elements should be written in full in the first instance

Introduction

Line 29 There is some repetition in this sentence which could be amended by deleting “and caused a widespread outbreak”

Line 68 “for and area” should be “for an area”

Line 75-79 This paragraph does not read well. Particularly “A virus is known…host for survival” does not fit with the other sentences. This paragraph might be rewritten and joined with the previous.

Methods

Study sites – can these sites be highlighted on the map in some way?

Table 1 – you could consider moving this table to the supplementary information?

Soil sampling – a more detailed explanation of “each master horizon” line 118 is required. At least what each one is named since this comes up later and are named in Supplementary Table 1

Line 133 KCl should be written out in full

Line 173 The sentence starts with a number 0.5ml this should be reworded

Line 190-202 are there any references for your approach to Discriminant analysis

Results

Line 215 Again I wonder if these cations and elements should be written in full in the first instance. And possibly mentioned in the methods?

Table 2 and 3 – please bold or highlight the significant variables

Fig 2 – I assume the numbers on the horizontal axis are the soil layers? I could not find this described in the caption or the text. Please add and also the names

Line 250 these four variables are not the same as described in the discussion line 282 and again are different to the three mentioned in the conclusion line 321 and in the abstract. Maybe this is due to changes between drafts that have not been reflected throughout the text???

Discussion

There is no comment on the relationship between Br and mortalities? But it is mentioned in the results (line 250) and the discussion (line 282) as a significant variable. There seems to be discrepancy between the results, the discussion and the conclusion and the abstract about which variables are significantly associated with RVF mortalities. The conflicting variables are Ca2+, Mg2+ and Br.

Reviewer #2: In the manuscript by Verster et al, the authors attempt to correlate variables found in soil to areas associated with Rift Valley fever virus (RVFV). As this virus is associated with increased rainfall, and is a well-known zoonotic agent transmitted by mosquitos, an attempt to associate factors such as soil minerality, as well as factors associated with soil moisture were measured. This involved comparing sites where RVFV have been known to exist, and comparing them to sites where RVFV has not been known to exist.

Interestingly, the conclusions that the authors propose seem to be at odds with other publications, where soil conditions of RVFV+ regions are considerably different than the ones proposed as ideal in this publication. This makes the reviewer somewhat skeptical of the practical findings of this publications.

While the reviewer is not terribly convinced of the overall findings of the paper, the methods used appear to be sound ; and thus, this paper should probably be accepted. However, one wonders whether this simply isn’t a case of « statistical artistry »--in other words, a case of presenting data in a manner that would make things more significant than they actually are.

Nevertheless, this paper should be considered for publication, assuming that the following concerns are addressed :

Minor concerns :

1) Don’t call it « the Rift Valley fever virus »….it’s just « Rift Valley fever virus »

2) Page 3, line 68—« ….for an area to be….. »

3) Page 4, line 76-77---please remove that sentence, as it makes no sense.

4) Page 7, line 119-120—«…and classification was done IUSS Working Group WRB »--what does that mean ?

6. PLOS authors have the option to publish the peer review history of their article (what does this mean?). If published, this will include your full peer review and any attached files.

Reviewer #1: No

Reviewer #2: No

---

## [Author Response · Author response to Decision Letter 0]

13 Dec 2019

Please accept herewith our revised submission (PONE-D-19-27989). The manuscript has been edited, taking the reviewer’s comments into account. We thank the reviewers for the positive and constructive suggestions that were made to improve the manuscript. We therefore believe that the current revision is of much better quality than the original. We have addressed most of the reviewer’s comments. Please find our detail comments below.

Reviewer #1:

1. Was done

2. We are of the opinion that element abbreviations are standard nomenclature end therefore do not need elucidation. However, we will abide by the editor’s decision in this regard.

3. We are of the opinion that element abbreviations are standard nomenclature end therefore do not need elucidation. However, we will abide by the editor’s decision in this regard.

4. Line 29: Was done

5. Line 68: Was done

6. Line 75-79: Was done

7. Study sites: Figure 1 has been edited to include the study sites.

8. Table 1: We decided to leave Table 1 in the main body text.

9. Soil sampling: The term “master horizon” is common knowledge for anyone working with soil, and it is referenced. We therefore did not change anything. The other reviewer also did not comment on this.

10. Line 133: We are of the opinion that element abbreviations are standard nomenclature end therefore do not need elucidation. However, we will abide by the editor’s decision in this regard.

11. Line 173: Was done

12. Line 190-202: Was done

13. Line 215: We are of the opinion that element abbreviations are standard nomenclature end therefore do not need elucidation. However, we will abide by the editor’s decision in this regard.

14. Table 2 & 3: Values differing significantly were highlighted.

15. Fig 2: This was done

16. Line 250: These sentences were edited to improve clarity.

17. Discussion: According to us the distinction is quite clear. There were eight variables (CEC, exchangeable Ca2+, exchangeable K+, exchangeable Mg2+, soluble Ca2+, medium sand, As, and Br) that differed significantly between the with and without RVFV. From these eight variable only four (exchangeable K+, exchangeable Mg2+, medium sand, and Br) were used in the discriminant function, since the inclusion of the other four would be superfluous. Also, the other reviewer did not comment on this.

Reviewer #2:

Reviewer 2 states: “Interestingly, the conclusions that the authors propose seem to be at odds with other publications, where soil conditions of RVFV+ regions are considerably different than the ones proposed as ideal in this publication.” We have found only two papers (Hightower et al., 2012; Sindato et al., 2016) relating soil properties to RVF. If anything, the findings of these papers support, rather than contradict our findings. We have added some sentences in the manuscript to clarify this.

1. Was done

2. Was done

3. Was done

4. This should be clear for a reader interested in soils; however, we added some words to explain it better.

Funding Disclosure:

The authors approve the following statements:

Financial Disclosure:

"WBK, MKR and JTP are thankful to the U.S. Department of Defense, Defense Threat Reduction Agency's Biological Threat Reduction Program for funding. Grant number: HDTRA1-14-1-0029. URL: https://www.dtra.mil/Mission/Mission-Directorates/Cooperative-Threat-Reduction/#. The funders had no role in study design, data collection and analysis, decision to publish, or preparation of the manuscript. The project depicted is sponsored by the U.S. Department of Defense, Defense Threat Reduction Agency. The content of the information does not necessarily reflect the position or the policy of the federal government, and no official endorsement should be inferred.

Claudia Cordel is affiliated with ExecuVet Veterinary Clinical and Scientific Consulting a private consulting company hired through the funding received by the listed funders.

ExecuVet Veterinary Clinical and Scientific Consulting provided support in the form of salaries for authors CC, but did not have any additional role in the study design, data collection and analysis, decision to publish, or preparation of the manuscript. The specific role of this author is articulated in the ‘author contributions’ section.”

Competing Interest:

"We have the following interests: Claudia Cordel is affiliated with ExecuVet Veterinary Clinical and Scientific Consulting. ExecuVet Veterinary Clinical and Scientific Consulting is a private company that was hired as a subcontractor to implement aspects of the research related to field data collection. They made no contributions toward funding this project.

There are no patents, products in development or marketed products to declare. This does not alter our adherence to all the PLOS ONE policies on sharing data and materials."

---

## [Decision Letter · Decision Letter 1]

4 Mar 2020

PONE-D-19-27989R1

Selected wetland soil properties correlate to Rift Valley fever livestock mortalities reported in 2009-10 in central South Africa

PLOS ONE

Dear Prof van Huyssteen,

Thank you for submitting your manuscript to PLOS ONE. After careful consideration, we feel that it has merit but does not fully meet PLOS ONE’s publication criteria as it currently stands. Therefore, we invite you to submit a revised version of the manuscript that addresses the points raised during the review process.

We would appreciate receiving your revised manuscript by Apr 18 2020 11:59PM. To enhance the reproducibility of your results, we recommend that if applicable you deposit your laboratory protocols in protocols.io, where a protocol can be assigned its own identifier (DOI) such that it can be cited independently in the future. For instructions see: http://journals.plos.org/plosone/s/submission-guidelines#loc-laboratory-protocols

We look forward to receiving your revised manuscript.

Kind regards,

Daehyun Kim, Ph.D.

Academic Editor

PLOS ONE

Additional Editor Comments (if provided):

Dear Authors,

Thank you for submitting a revised version of your work. I have now received comments from two reviewers who had evaluated the original version of this manuscript. Please do note that both still have some important concerns about this revision. In particular, they seem to feel bad to see how you disrespectfully responded to their comments. Keep in mind that the referees sacrificed their valuable time to help you improve your research. If you decided to submit a R2 to PLOS ONE, I would like you to do your best to show your gratitude and respect to the reviewers because the same two people will be invited again. Also, below lists a number of issues of mine that should be addressed in your R2:

(1) The Intro section should be improved. From the beginning to Line 54, you provided sort of textbook knowledge only. Please write a more argumentative introduction, rather than an explanative one. Clearly discuss what knowledge gaps exist, why these gaps are important, what your hypothesis is, etc.

(2) Line 118 -- It is very easy to follow Reviewer 1 and to provide a very simple description of "each master horizon." Please do so, rather than saying "everybody knows that."

(3) The section titles "Physical and chemical analysis," "Mineralogical analysis," "Soil microbiology analysis," and "Active carbon analysis" could be combined to a single one, titled, for example "Soil analysis."

(4) Line 186 -- Explain the van der Waerden analysis.

(5) Lines 284-285 -- Why don't you present the results of pairwise correlation analysis between soil properties?

Reviewers' comments:

Reviewer's Responses to Questions

**Comments to the Author**

1. If the authors have adequately addressed your comments raised in a previous round of review and you feel that this manuscript is now acceptable for publication, you may indicate that here to bypass the “Comments to the Author” section, enter your conflict of interest statement in the “Confidential to Editor” section, and submit your "Accept" recommendation.

Reviewer #1: (No Response)

Reviewer #2: (No Response)

2. Is the manuscript technically sound, and do the data support the conclusions?

Reviewer #1: Yes

Reviewer #2: Yes

3. Has the statistical analysis been performed appropriately and rigorously? 

Reviewer #1: Yes

Reviewer #2: I Don't Know

4. Have the authors made all data underlying the findings in their manuscript fully available?

Reviewer #1: Yes

Reviewer #2: Yes

5. Is the manuscript presented in an intelligible fashion and written in standard English?

Reviewer #1: Yes

Reviewer #2: Yes

6. Review Comments to the Author

Reviewer #1: Many thanks to the authors for making the minor amendments I recommended in the first review.

However, there is still one issue remaining and the authors may not have understood my previous comments. I have now highlighted this as a major issue which needs to be rectified before publication.

Major amendment

There is a discrepancy between the described results and the discussion with different variables being described as being included in the final model. (Line numbers refer to the marked-up version)

On line 255 of the marked-up version the results clearly state the following “four variables (exchangeable K+, exchangeable Mg2+, medium sand, and Br) were identified by the stepwise variable selection of the discriminant analysis to discriminate between sites with and without reported RVF mortality”. There is a clear list of FOUR variables.

However, in the discussion line 275 the authors refer to three variables “soluble Ca2+ and exchangeable K+, and medium sand”. The authors are discussing different variables from their results – soluble Ca2+ has been added and Br and exchangeable Mg2+ dropped?

Then again on line 290 they refer to four variables being in the model “Although only soluble Ca2+, exchangeable K+, Br and medium sand were used in the final model”. Again a different list of variables. Please note that soluble Ca2+ is not described in the results in page 15 as being in the final model.

In the conclusion line 333-335 the authors again refer to three variables “Three soil characteristics (soluble Ca2+, exchangeable K+, and medium sand) were subsequently selected for inclusion into a discriminant function that could potentially be used to predict an RVF outbreak site”

In the abstract line 11-12 the authors write “Three soil characteristics (soluble Ca 2+, exchangeable K +, and medium sand) were consequently included in a discriminant function that could potentially be used to predict a Rift Valley fever outbreak site”.

This inconsistency regarding the number of variables and the list of variables included in the final model NEEDS to be rectified before publication.

Minor amendments

The authors might colour the sampling sites more clearly in Figure 1

Line 172 This sentence starts with a digit “2.5g” and should be rewritten

Line 178 measureing is misspelled

Line 297 and Line 298 there appears to be issues with the references?

Reviewer #2: The Reviewer only has two points to make:

1) The submission has been made to PLoS One---not the Journal of Soil Sciences. Therefore, the authors' assertion that certain terminology "is common knowledge for anyone working with soil" is inappropriate. Comments like these generally don't help your argument; and to be honest, the Reviewer finds it to be quite rude. The authors should appreciate that comments like these may automatically get your manuscript rejected.

2) Just because the second Reviewer does't repeat a concern that the first Reviewer raises, it doesn't mean that the first Reviewer's concerns are not justified. As the second reviewer, I actually agree with Reviewer #1 on most points (including points 9 and 17); and just for your information, Reviewer # 2 also had to look up the definition of "Master Horizon". This is simply another example of poor conduct on the part of the authors.

7. PLOS authors have the option to publish the peer review history of their article (what does this mean?). If published, this will include your full peer review and any attached files.

Reviewer #1: No

Reviewer #2: No

---

## [Author Response · Author response to Decision Letter 1]

26 Mar 2020

I have included a separate letter in this regard, but also copy it here:

Editor:

(1) The Intro section should be improved. From the beginning to Line 54, you provided sort of textbook knowledge only. Please write a more argumentative introduction, rather than an explanative one. Clearly discuss what knowledge gaps exist, why these gaps are important, what your hypothesis is, etc.

This was done.

(2) Line 118 -- It is very easy to follow Reviewer 1 and to provide a very simple description of "each master horizon." Please do so, rather than saying "everybody knows that."

This was done.

(3) The section titles "Physical and chemical analysis," "Mineralogical analysis," "Soil microbiology analysis," and "Active carbon analysis" could be combined to a single one, titled, for example "Soil analysis."

This was done.

(4) Line 186 -- Explain the van der Waerden analysis.

This was done.

(5) Lines 284-285 -- Why don't you present the results of pairwise correlation analysis between soil properties?

This was done. (Table 4)

Reviewer #1:

There is a discrepancy between the described results and the discussion with different variables being described as being included in the final model. (Line numbers refer to the marked-up version). On line 255 of the marked-up version the results clearly state the following “four variables (exchangeable K+, exchangeable Mg2+, medium sand, and Br) were identified by the stepwise variable selection of the discriminant analysis to discriminate between sites with and without reported RVF mortality”. There is a clear list of FOUR variables. However, in the discussion line 275 the authors refer to three variables “soluble Ca2+ and exchangeable K+, and medium sand”. The authors are discussing different variables from their results – soluble Ca2+ has been added and Br and exchangeable Mg2+ dropped? Then again on line 290 they refer to four variables being in the model “Although only soluble Ca2+, exchangeable K+, Br and medium sand were used in the final model”. Again a different list of variables. Please note that soluble Ca2+ is not described in the results in page 15 as being in the final model. In the conclusion line 333-335 the authors again refer to three variables “Three soil characteristics (soluble Ca2+, exchangeable K+, and medium sand) were subsequently selected for inclusion into a discriminant function that could potentially be used to predict an RVF outbreak site” In the abstract line 11-12 the authors write “Three soil characteristics (soluble Ca2+, exchangeable K+, and medium sand) were consequently included in a discriminant function that could potentially be used to predict a Rift Valley fever outbreak site”. This inconsistency regarding the number of variables and the list of variables included in the final model NEEDS to be rectified before publication.

This was corrected.

Minor amendments

The authors might colour the sampling sites more clearly in Figure 1

I have used un-filled triangles to indicate the study sites, because they overlap at the scale of presentation. I therefore did not change the current layout, although I can easily do so if required.

Line 172 This sentence starts with a digit “2.5g” and should be rewritten

This was corrected.

Line 178 measureing is misspelled

This was corrected.

Line 297 and Line 298 there appears to be issues with the references?

This was corrected. I have also checked the other references, since more such issues were found. It seems that the cross references went awry during the editing and by using track changes.

Reviewer #2:

1) The submission has been made to PLoS One---not the Journal of Soil Sciences. Therefore, the authors' assertion that certain terminology "is common knowledge for anyone working with soil" is inappropriate. Comments like these generally don't help your argument; and to be honest, the Reviewer finds it to be quite rude. The authors should appreciate that comments like these may automatically get your manuscript rejected.

I hereby offer my sincere, unreserved apology and assure you that no affront was intended.

2) Just because the second Reviewer does't repeat a concern that the first Reviewer raises, it doesn't mean that the first Reviewer's concerns are not justified. As the second reviewer, I actually agree with Reviewer #1 on most points (including points 9 and 17); and just for your information, Reviewer # 2 also had to look up the definition of "Master Horizon". This is simply another example of poor conduct on the part of the authors.

We will gladly include any elucidation required, but there is also an assumed common discourse, which in this case obviously did not hold true. Once again, my sincere apologies.

---

## [Decision Letter · Decision Letter 2]

12 Apr 2020

PONE-D-19-27989R2

Selected wetland soil properties correlate to Rift Valley fever livestock mortalities reported in 2009-10 in central South Africa

PLOS ONE

Dear Prof van Huyssteen,

Thank you for submitting your manuscript to PLOS ONE. After careful consideration, we feel that it has merit but does not fully meet PLOS ONE’s publication criteria as it currently stands. Therefore, we invite you to submit a revised version of the manuscript that addresses the points raised during the review process.

We would appreciate receiving your revised manuscript by May 27 2020 11:59PM. To enhance the reproducibility of your results, we recommend that if applicable you deposit your laboratory protocols in protocols.io, where a protocol can be assigned its own identifier (DOI) such that it can be cited independently in the future. For instructions see: http://journals.plos.org/plosone/s/submission-guidelines#loc-laboratory-protocols

We look forward to receiving your revised manuscript.

Kind regards,

Daehyun Kim, Ph.D.

Academic Editor

PLOS ONE

Additional Editor Comments (if provided):

Dear Authors,

I think that the review process for this manuscript is almost done. The two reviewers I invited seem to be very satisfied with this second revised version, and I concur with them. Because one of the reviewers still has a few small concerns, I hereby request that you address them in your R3. Thank you.

Sincerely,

Daehyun Kim

Academic Editor, Ecology section

Reviewers' comments:

Reviewer's Responses to Questions

**Comments to the Author**

1. If the authors have adequately addressed your comments raised in a previous round of review and you feel that this manuscript is now acceptable for publication, you may indicate that here to bypass the “Comments to the Author” section, enter your conflict of interest statement in the “Confidential to Editor” section, and submit your "Accept" recommendation.

Reviewer #1: (No Response)

Reviewer #2: All comments have been addressed

2. Is the manuscript technically sound, and do the data support the conclusions?

Reviewer #1: Yes

Reviewer #2: Yes

3. Has the statistical analysis been performed appropriately and rigorously? 

Reviewer #1: Yes

Reviewer #2: Yes

4. Have the authors made all data underlying the findings in their manuscript fully available?

Reviewer #1: Yes

Reviewer #2: Yes

5. Is the manuscript presented in an intelligible fashion and written in standard English?

Reviewer #1: Yes

Reviewer #2: Yes

6. Review Comments to the Author

Reviewer #1: Dear Authors,

Thank you for making the recommended revisions. It was a pleasure to read your manuscript.

There are a couple of typographical errors

Line 54 of the unmarked version "with with".

Line 86 of the unmarked version in the Table 1 Heading "World Reference Base" not Bace

Line 159 of the unmarked version I wanted to check the nomenclature the authors have written "B/B1/G/C1" should the G here be a C? and the same for line 217 the Figure 2 label

Otherwise I have no further comments.

Warm regards

Reviewer #2: (No Response)

7. PLOS authors have the option to publish the peer review history of their article (what does this mean?). If published, this will include your full peer review and any attached files.

Reviewer #1: No

Reviewer #2: No

---

## [Author Response · Author response to Decision Letter 2]

14 Apr 2020

We have made two corrections, but declined the third. Details are given in the response to reviewers' letter.

---

## [Editor Report · Decision Letter 3]

16 Apr 2020

Selected wetland soil properties correlate to Rift Valley fever livestock mortalities reported in 2009-10 in central South Africa

PONE-D-19-27989R3

Dear Dr. van Huyssteen,

We are pleased to inform you that your manuscript has been judged scientifically suitable for publication and will be formally accepted for publication once it complies with all outstanding technical requirements.

With kind regards,

Daehyun Kim, Ph.D.

Academic Editor

PLOS ONE
---

## [Editor Report · Acceptance letter]

6 May 2020

PONE-D-19-27989R3 

Selected wetland soil properties correlate to Rift Valley fever livestock mortalities reported in 2009-10 in central South Africa 

Dear Dr. van Huyssteen:

I am pleased to inform you that your manuscript has been deemed suitable for publication in PLOS ONE. Congratulations! Your manuscript is now with our production department. 

With kind regards,

on behalf of

Dr. Daehyun Kim 

Academic Editor

PLOS ONE